# Fibrinogen Alpha Chain as a Potential Serum Biomarker for Predicting Response to Cisplatin and Gemcitabine Doublet Chemotherapy in Lung Adenocarcinoma: Integrative Transcriptome and Proteome Analyses

**DOI:** 10.3390/ijms26031010

**Published:** 2025-01-24

**Authors:** Pritsana Raungrut, Jirapon Jirapongsak, Suchanan Tanyapattrapong, Thitaya Bunsong, Thidarat Ruklert, Kannika Kueakool, Paramee Thongsuksai, Narongwit Nakwan

**Affiliations:** 1Division of Biomedical Sciences and Biomedical Engineering, Faculty of Medicine, Prince of Songkla University, Hat Yai 90112, Songkhla, Thailand; jirapon.jpk@gmail.com (J.J.); bsuchanant@gmail.com (S.T.); 2Division of Pulmonology, Department of Medicine, Hat Yai Medical Education Center, Hat Yai Hospital, Hat Yai 90112, Songkhla, Thailand; darryqueenz@gmail.com (T.B.); thidaratruklerd@gmail.com (T.R.); 3Faculty of Medicine, Prince of Songkla University, Hat Yai 90112, Songkhla, Thailand; kannika.wii@gmail.com; 4Department of Pathology, Faculty of Medicine, Prince of Songkla University, Hat Yai 90112, Songkhla, Thailand; tparamee@gmail.com

**Keywords:** fibrinogen alpha chain, lung adenocarcinoma, transcriptome, proteome, chemotherapy

## Abstract

Cisplatin combined with gemcitabine, a doublet regimen, is the first-line treatment for patients with advanced lung adenocarcinoma (ADC); however, the treatment response remains poor. This study aimed to identify potential biomarkers for predicting response to cisplatin and gemcitabine. Tissue transcriptome and blood proteome analyses were conducted on 27 patients with lung ADC. Blood-derived proteins that reflected tissue-specific biomarkers were obtained using Venn diagrams. The candidate proteins were validated by Western blotting. Lentivirus-mediated short hairpin RNA interference was used to verify the functional roles of the candidate proteins in human A549 cells. We identified 417 differentially expressed genes, including 52 upregulated and 365 downregulated genes, and 31 differentially expressed proteins, including 26 upregulated and 5 downregulated proteins. Integrative analysis revealed the presence of alpha-1-acid glycoprotein 1 (A1AG1) and fibrinogen alpha chain (FGA or FIBA) in both the tissue and serum. FGA levels were elevated in responders compared to non-responders, and reduced serum FGA levels were correlated with resistance to this regimen. Moreover, FGA knockdown in A549 cells resulted in resistance to the doublet regimen. Our findings indicate that FGA is a tissue-specific serum protein that may function as a blood-based biomarker to predict the response of patients with lung ADC to cisplatin plus gemcitabine chemotherapy.

## 1. Introduction

Lung cancer is the most frequently diagnosed cancer, accounting for 12.4% of all cancer sites, and was expected to be the leading cause of cancer-related death (18.7% of all cancer sites) worldwide by 2022 [1]. In 2020, lung adenocarcinoma (ADC) was a frequent histological subtype of non-small cell lung cancer (NSCLC), accounting for approximately 39% and 57% of all male and female patients with lung cancer, respectively [2,3]. More than 70% of patients are at an advanced stage at the time of diagnosis, and only 8.4% of patients with lung ADC remain alive after 5 years [4]. Although numerous targeted and immunotherapeutic agents are employed in treating advanced lung ADC, chemotherapy remains a vital part of treatment. Chemotherapy is cost-effective, and most patients with advanced lung ADC lack mutations that would enable them to be treated with targeted treatment. Doublet chemotherapy, which combines a platinum-based agent (cisplatin or carboplatin) with a third-generation agent, is the first-line treatment for advanced lung ADC [5]. A platinum-based agent with an antimetabolite (gemcitabine and pemetrexed) is a frequently used regimen. Although this doublet regimen may be more cost-effective and convenient than other doublet regimens [6], a response rate of <30% has been reported [7,8]. Consequently, it is crucial to identify predictive biomarkers that could assist in selecting suitable regimens.

The emergence of “omics” technologies has changed the field of cancer over the previous few decades. Transcriptomics (the study of a set of expressed genes) and proteomics (the study of a set of expressed proteins) are the two most frequently used tools for identifying and validating potential biomarkers [9]. Transcriptome analysis showed 4 genes as potential targets for the carboplatin plus paclitaxel regimen in endometrial cancer cells [10], whereas 15 genes were associated with cisplatin resistance in lung cancer cells [11]. Similarly, proteomic analyses have indicated that several blood proteins can predict responses to cisplatin plus pemetrexed [12] and cisplatin plus docetaxel regimens in patients with NSCLC [13].

Although advances in the gene expression profiling of tumor cells or tissues have led to the discovery of new biomarkers for therapeutic response, collecting specimens in sufficient quantities often presents challenges. Furthermore, blood-based protein biomarkers may be useful because of their accessibility; however, they may not accurately reflect tumor-specific biomarkers. This study identified potential tissue-specific blood proteins for predicting the response to a doublet cisplatin and gemcitabine regimen through an integrated analysis of tissue transcriptome and blood proteome data from patients with lung ADC. Subsequently, we conducted a functional study of the potential proteins in lung ADC cell lines using lentivirus-mediated short-hairpin (sh)RNA interference. This study offers new insights for predicting the chemotherapy response of lung ADC.

## 2. Results

### 2.1. Study Design and Patient Characteristics

Figure 1 presents a flowchart of the study. Tumor tissues were collected for transcriptome sequencing, and matching sera were collected for proteome profiling. Following gene identification, the differentially expressed genes (DEGs) and differentially expressed proteins (DEPs) were identified. An integrated analysis of the DEGs and DEPs was conducted to assess whether the encoded proteins of the DEGs could be detected in the blood. Overlapping proteins were verified using Western blotting. Finally, a candidate protein was selected for functional investigation using shRNA interface.

A total of 27 patients, 14 men and 13 women, were treated with a combination of gemcitabine and cisplatin chemotherapy. Their median age was 60.0 years (range: 39–84 years). Computed tomography of the chest was performed before and after treatment, at the third cycle, and onwards to assess the response to the therapy according to the Response Evaluation Criteria in Solid Tumors (RECIST) [14]. Among the 27 patients, 17 (6 SD cases and 11 PD cases) were non-responders, and 10 (1 CR case and 9 PR cases) were responders (Table 1). For transcriptome sequencing, we found that only four samples achieved the minimum requirements for both yield (≥1000 ng) and quality (RIN ≥ 5) (Appendix A). Because of the insufficient quality of RNA in our tissue samples, we obtained transcriptome sequencing data from TCGA database. Fourteen tissue-based transcriptome datasets were generated (Appendix A). The patients had a median age of 61.50 years (range: 49–77 years). All patients diagnosed with advanced-stage (III or IV) ADC (seven men and seven women) were treated with a combination of either carboplatin or cisplatin, along with either gemcitabine or pemetrexed. There were eight responders, including six CR cases and two PR cases. Additionally, there were six non-responders, of whom four had SD and two had PD. For proteomic profiling, 12 serum samples (responders = 6 and non-responders = 6) were chosen based on sex-age matching. Two serum samples in each group were pooled. As a result, three pooled samples from each group were used in duplicate.

### 2.2. DEGs Between Responders and Non-Responders

Using our 4 established sequences (responders = 1, R1 and non-responders = 3, NR 1-NR3) and 14 transcriptome sequences from TCGA database (responders = 8, TCGA-R1–TCGA-R8 and non-responders = 6, TCGA-NR1–TCGA-NR6), a heat map analysis showed two distinct gene expression patterns between the responders and non-responders (Figure 2A). A total of 417 DEGs were identified, 52 of which were upregulated and 365 of which were downregulated in responders compared with non-responders (Figure 2B). K-means clustering generated 27 clusters, with the largest containing 27 genes and the smallest containing 2 genes (Figure 2C). Three main clusters were identified in this study. Cluster A comprised 27 genes enriched in the complement and coagulation cascades and protein-lipid complexes. Clusters B and C each consisted of six genes associated with regionalization, interneuron differentiation, and the suppression of voltage-gated channels. The most enriched GO terms are shown in Figure 2D; these genes were mainly found in cellular components, specifically in the plasma membrane. Regarding molecular function, the most significantly enriched gene was associated with clathrin binding. Reactome pathway analysis revealed the significant enrichment of genes related to biological oxidation. CALCA, TRPC7, AKR1B10, CPS1, and ALDH3A1 were the most significantly upregulated DEGs, whereas NOXO1, NCR2, PRSS54, LHFPL3, and NR1H2 were the most significantly downregulated DEGs (Table 2).

### 2.3. DEPs Between Responders and Non-Responders

From our six pools of twelve serum samples, including three pooled responders in duplicate (PS1 #1, PS1 #2, PS2 #1, PS2 #2, PS3 #1 and PS3 #2) and three pooled non-responders in duplicate (PS4 #1, PS4 #2, PS5 #1, PS5 #2, PS6 #1 and PS6 #2), a heat map revealed 31 DEPs in the sera of responders compared to non-responders (Figure 3A). Of these DEPs, 26 were upregulated and 5 were downregulated in responders compared to non-responders (Figure 3B). The K-means clustering generated three clusters (Figure 3C). Cluster A comprised 13 genes and was enriched in the complement and coagulation cascades and protein–lipid complexes. Cluster B contained eight genes associated with complement activation and the classical pathway. Cluster C was very small, with only three genes identified as being linked to nitric oxide transport. The acute-phase response, blood microparticles, and serine-type endopeptidases were dominant, with the highest significance indicated for biological processes, cellular components, and molecular functions, respectively. Reactome enrichment analysis demonstrated that these DEPs were associated with platelet activation, as shown in Figure 3D. The most significantly upregulated DEPs were SAA2, RET4, LV147, K2C1, and HBA. In contrast, the downregulated DEPs were KAIN, A1AG2, HRG, HSBG and IGHM (Table 3).

### 2.4. Identification of Blood-Secretory Proteins via Integrative Analysis and Validation

The results showed that 408 DEGs were annotated to known proteins, whereas 9 DEGs remained unidentified. Using a Venn diagram, we discovered that A1AG1 and FGA were predicted to be proteins secreted into the blood (Figure 4A and Appendix A). A1AG1 and FGA were expressed at higher levels in the responders than in the non-responders. To verify differential expressions, the level of expression was confirmed in the sera of 27 patients by immunoblotting (responders = 10 and non-responders = 17). Serum A1AG1 expression was higher in the responders (mean: 0.735; SEM: 0.077) than in the non-responders (mean: 0.447; SEM: 0.066). However, there was no statistically significant difference in the expression between the two groups (Figure 4B,D). Responders showed significantly higher expression levels of FGA (mean: 0.682, SEM: 0.131) than non-responders (mean: 0.447, SEM: 0.066), with a *p*-value of 0.047 (Figure 4C,E).

### 2.5. Association of A1AG1 and FGA with Clinicopathological Variables

The serum level of A1AG1 and FIBA proteins, as determined by Western blotting, established cut-off values of ≤0.424 for A1AG1 and ≤0.776 for FGA at the lower value, and >0.424 for A1AG1 and >0.776 for FGA at the upper value. The associations between clinicopathological variables and protein expression are shown in Table 4. High A1AG1 expression was significantly associated with the response to the combination of gemcitabine and cisplatin treatment (*p* = 0.018), whereas low FIBA expression was associated with the response to this regimen (*p* = 0.029).

### 2.6. FGA Knockdown Constructed by Lentivirus-Mediated shRNA Infection of A549 Cells

Stable cell lines were constructed with knockdown of FGA in the A549 cells to further investigate the role of FGA in the chemotherapy response. GFP expression in shRNA-FGA cells (Figure 5B) confirmed the successful transduction of the lentivirus into A549 cells, in contrast to the absence of GFP expression in parental A549 cells (Figure 5A). The expression of FGA was substantially decreased by approximately 70–80% in shRNA-FGA relative to that in A549 cells (Figure 5C,D; *p* < 0.001). The number of viable A549 cells decreased time-dependently following drug treatment. No reduction in the number of viable shRNA-FGA cells was observed (Figure 5E).

The dose–response curves revealed that shRNA-FGA cells exhibited increased resistance to a combination of cisplatin and gemcitabine by approximately five-fold (IC_50_ = 7.70 µM for shRNA-FGA versus IC_50_ = 1.47 µM for A549) (Figure 6A) and to gemcitabine alone by approximately two-fold (IC_50_ = 1.65 µM for shRNA-FGA versus IC_50_ = 0.80 µM for A549) (Figure 6B). In contrast, decreased FGA levels in shRNA-FGA cells did not confer resistance to cisplatin alone (Figure 6C). Treatment of A549 cells with doublet chemotherapy also induced the expression of cleaved forms of PARP and caspase-3 and -7, with no significant difference compared to their total forms. In contrast, shRNA-FGA cells treated with doublet chemotherapy did not show PARP cleavage and caspase-3/-7 activation, with a significant change in the level of all cleaved forms compared to their total forms (*p* < 0.05 (Figure 6D–G). The wound healing rate of shRNA-FGA cells was significantly higher than that of A549 cells (*p* = 0.008) (Figure 6H). After 3 days, the rate of wound healing increased by 80% in shRNA-FGA cells, whereas A549 cells displayed only a 30% increase in the wound healing rate. These findings indicate that reduced FGA expression increases resistance to a combination of cisplatin and gemcitabine and promotes cell migration in lung cancer cells.

## 3. Discussion

In this study, we detected A1AG1 and FGA in the tissues and sera of patients with lung ADC receiving a combination of cisplatin and gemcitabine. We also found that the serum FGA level was significantly higher in responders than in non-responders, indicating that it was associated with chemotherapy response. In addition, FGA knockdown with lentivirus-mediated shRNA interference enhanced the resistance of lung cancer cells to cisplatin and gemcitabine.

Alpha-1-acid glycoprotein 1 (A1AG1), also known as orosomucoid 1 (ORM1) or AGP1, is encoded by the ORM1 gene. It is a glycosylated protein found in the bloodstream that has been shown to have immunomodulatory and anti-inflammatory properties [15]. Elevated A1AG1 levels have been found in the sera of patients with hepatocellular carcinoma [16], laryngeal cancer [17], pancreatic ductal ADC [18], and lung cancer [19] compared to non-cancerous patients or healthy persons. Some of these proteins have demonstrated clinical value as diagnostic biomarkers [16,18]. Hyung et al. revealed that six blood biomarkers, including ORM1, predicted the response of patients with breast cancer to a combination of docetaxel and doxorubicin [20]. Zhou et al. reported that high serum levels of A1AG1 correlated with no response in patients with extranodal NK/T-cell lymphoma after treatment with a combination of pegaspargase and gemcitabine [21]. Our previous study aligns with a later study indicating that A1AG1 levels were markedly increased in non-responders with squamous cell carcinoma (SCC) but not in ADC receiving carboplatin and paclitaxel [22]. In the current study, we revealed an inconsistent finding, demonstrating that serum A1AG1 levels were higher in responders than non-responders; however, this difference was not statistically significant. This increased level was associated with response to doublet cisplatin and gemcitabine in patients with lung ADC. These findings suggest that the serum A1AG1 level may serve as a predictive response to platinum-based doublet chemotherapy, notably for either carboplatin plus paclitaxel or cisplatin plus gemcitabine regimens; however, its applicability as a blood-based biomarker is dependent on the histological subtype of NSCLC.

Fibrinogen alpha chain (FGA), also known as FIBA, is a subunit of fibrinogen that consists of three unique pairs of polypeptide chains, alpha, beta, and gamma, linked by disulfide bonds to form a symmetric dimeric structure [23]. It is a vital protein that is involved in the coagulation of blood in humans [23]. This was supported by our findings that FGA was enriched in the complement and coagulation cascades, as determined by cluster analysis of both DEGs and DEPs. FGA has been detected in the urine for the diagnosis and prognosis of urinary tract infections [24] and bladder cancer [25]. Furthermore, FGA was found to be decreased in gastric cancer tissues and cell lines [26,27]; its low expression is associated with a favorable prognosis [26]. Several studies using serum proteomics have indicated that protein peaks of FGA exhibited elevated expression in gastric ADC [28], stage I lung SCC [29], and colorectal cancer [30]. Furthermore, combining FGA with other proteins or FGA alone can serve as diagnostic or prognostic biomarkers [28,29,30]. Li et al. revealed that FGA knockdown in endometrial stromal cells using lentiviral shRNA interference also reduced FGA protein levels in the conditioned media. Moreover, conditioned media from endometrial stromal cells with FGA knockdown may suppress the angiogenic ability of endothelial cells via VEGFR2–FAK signaling [31]. Some studies have indicated that FGA knockout via CRISPR/Cas9 genome editing or FGA knockdown via shRNA interference leads to enhanced cell proliferation, migration, invasion, and metastasis of lung ADC cells via the integrin–AKT signaling pathway [32], and in gastric cancer cells via modulation of the FAK–ERK pathway [27]. These results confirmed that FGA is a secreted protein associated with angiogenesis and cancer progression.

Our recent findings support those of previous studies, indicating the presence of FGA in both the tissues and sera of patients with lung ADC. Furthermore, we revealed that reduced serum FGA levels correlated with a lack of response to cisplatin and gemcitabine regimens in patients with lung ADC. This result was confirmed by the in vitro results; we showed that resistance to this doublet treatment in lung ADC cells developed following a reduction in FGA expression by lentivirus-mediated shRNA via inhibition of the apoptosis pathway. Nevertheless, the current investigation was constrained by the restricted number of clinical samples, including both tissues and sera, which may hinder the assessment of putative biomarkers for FGA. Consequently, further clinical studies with larger sample sizes are required to confirm the role of FGA as a predictive biomarker in this context.

## 4. Materials and Methods

### 4.1. Subject Selection

Tumor tissues and corresponding serum samples were obtained from patients diagnosed with lung ADC between 2021 and 2023. The following inclusion criteria were applied: (1) a recent diagnosis of pathologically verified stage III or IV cancer, and (2) five or six cycles of first-line chemotherapy using a platinum agent in conjunction with an antimetabolite agent. Patients who had previously received any treatment for lung cancer and those who were lost to follow-up or died during treatment were excluded from the study. The study protocol was approved by the Human Research Ethics Committee of the Hat-Yai Hospital in Songkhla, Thailand. Written informed consent was obtained from all participants.

### 4.2. Collection of Tissue-Based Transcriptome Data from the Cancer Genome Atlas (TCGA)

The transcriptome data were retrieved from TCGA. Datasets were selected according to the following criteria: (1) tissue samples of patients with NSCLC; (2) advanced stage lung ADC (III or IV); (3) patients treated with a combination of platinum and antimetabolite agents; and (4) response to chemotherapy reported according to the RECIST [14]. Clinical information, including age, sex, stage, treatment, and response, was downloaded from the Genome Data General Database (GDC) data portal (https://portal.gdc.cancer.gov/projects/TCGA-LUAD (accessed on 7 June 2021)). 

### 4.3. Sample Preparation, Treatment, and Clinical Follow-Up

Lung cancer tissues were obtained under direct vision via bronchoscopy or transthoracic needle biopsy. Samples were immediately snap frozen in liquid nitrogen and then stored at −80 °C until RNA extraction. Blood samples were collected at the time of diagnosis, coagulated for 30 min at room temperature, centrifugation at 3400× *g* for 10 min, and the serum was collected. The serum was filtered through a polyvinylidene difluoride syringe filter with a pore size of 0.22 mm (Merck Millipore, Darmstadt, Germany), and aliquots were kept at −80 °C until use. According to the RECIST, the evaluated response was categorized into 4 groups: complete response (CR), defined as the disappearance of all target lesions; partial response (PR), involving a reduction of at least 30% from the total target lesions; stable disease (SD), when neither lesion reduction nor progression was present; and progressive disease (PD), when at least 20% progression of the target lesions or the presence of new lesions was observed. Responders were patients who exhibited either CR or PR, whereas non-responders were those with SD or PD.

### 4.4. Transcriptome Sequencing

Snap-frozen tissue was rapidly transferred to a new tube with 600 µL of buffer RLT Plus (Qiagen, Valencia, CA, USA) containing 1% β-mercaptoethanol (Sigma-Aldrich, St Louis, MO, USA). The samples were homogenized using microtube pestles. RNA was extracted using the RNeasy Plus Mini Kit (Qiagen) following the manufacturer’s instructions. The concentration and purity of the RNA was assessed using a NanoDrop^®^ ND-1000 UV-Vis spectrophotometer (Thermo Scientific, Waltham, MA, USA). RNA quality was measured using a 2100 Bioanalyzer (Agilent Technologies, Santa Clara, CA, USA); this provides a score (0–10) based on the RNA electrophoresis trace, known as the RNA integrity number (RIN). The RNA yield was considered adequate for transcriptome sequencing if 1000 ng of RNA was obtained. RNA quality was considered acceptable if the RIN was equal to or greater than 5. Total RNA was shipped to Macrogen Inc. (Seoul, Republic of Korea) on dry ice.

Transcriptomic sequencing was performed using a TruSeq mRNA library preparation kit (Illumina Inc., San Diego, CA, USA) and a Novaseq 6000 instrument (Illumina Inc.), according to the manufacturer’s instructions. Briefly, mRNA was isolated using poly A selection and chemically fragmented. The fragmented RNA was reverse transcribed into single-stranded cDNA using random hexamer priming, followed by double-stranded cDNA synthesis using TruSeq library construction. Short cDNAs were ligated to adapters and enriched using polymerase chain reaction (PCR) to generate the final cDNA library. Following amplification, RNA sequencing was performed using a NovaSeq 6000 instrument (Illumina, Inc. San Diego, CA, USA) with 150 bp paired-end sequencing according to the standard protocol.

### 4.5. Gene Identification and Data Analysis

Quality analysis of the raw transcriptome sequencing data was performed using the FastQC program version 0.11.9 (https://www.bioinformatics.babraham.ac.uk/projects/fastqc/ (accessed on 30 June 2021)). The sequences were trimmed to remove low-quality bases (<Q30) and adaptors using the Trimmomatic program version 0.32 (http://www.usadellab.org/cms/?page=trimmomatic (accessed on 30 June 2021)). High-quality clean reads were aligned to the reference human genome using STAR version 2.7.0a (https://github.com/alexdobin/STAR (accessed on 30 June 2021)). RSEM v1.2.12 software (https://github.com/deweylab/RSEM (accessed on 22 July 2021)) was used to estimate the raw read counts. The read count for each gene was determined as fragments per kilobase of exon per million fragments mapped (FPKM) using Cufflinks software version 2.2. (http://cole-trapnell-lab.github.io/cufflinks/manual/ (accessed on 22 July 2021)). DEGs between responders and non-responders were analyzed using the DESeq2 package in R (version 3.1.0) [33]. *p*-values were corrected for multiple testing using the False Discovery Rate (FDR) approach, and genes with adjusted *p*-values < 0.05 were considered significant. Log2 fold change ≥2 and an adjusted *p*-value < 0.05 were used as the cut-offs for DEGs.

### 4.6. Serum Proteomics

The serum was subjected to depletion using the Human 14 Multiple Affinity Removal System Column with a 4.6 × 100 mm dimension (Agilent Technologies, Inc., Santa Clara, CA, USA). The protein concentration of the depleted serum was determined using the Bradford assay (Bio-Rad Laboratories, Hercules, CA, USA). A 5-µg aliquot of depleted serum proteins for each pool was resuspended in 50 mM ammonium bicarbonate (Fluka Chemical Corp., Ronkonkoma, NY, USA). Pooled samples were reduced with dithiothreitol (Thermo Scientific) followed by alkylation with iodoacetamide (Sigma–Aldrich) in the dark for 30 min at room temperature. Trypsin (Thermo Fisher Scientific. Waltham, MA, USA) at an enzyme/substrate ratio of 1:25 (*w*/*w*) was added to the samples and incubated at 37 °C overnight. The digested samples were cleaned using ZipTip C18 pipette tips (Merck Millipore). Formic acid (1% final concentration; Sigma–Aldrich) was added, and samples were dried using a SpeedVac Evaporator (Labconco Corp., Kansas, MO, USA) prior to liquid chromatography coupled with tandem mass spectrometry (LC-MS/MS).

### 4.7. Protein Identification and Data Analysis

LC-MS/MS was performed as previously described [22]. Briefly, the samples were analyzed with the Thermo Scientific™ Dionex™ Ultimate™ 3000 RSLC Nano System coupled with electrospray ionization-ion trap tandem MS (Bruker Corp., Billerica, MA, USA) with a captive-electrospray ion source. Peptides were loaded on an Acclaim PepMapTM RSLC C18 column (150 mm × 75 μm dimension, 2 μm, 100 Å pore size) connected with an Acclaim PepMap100 C18 micro precolumn (5 mm × 300 μm dimension) (Thermo Fisher Scientific). A gradient of 0.1% formic acid in acetonitrile, 1–50% at 0–70 min, increased to 90% at 70–75 min, maintained at 90% for 5 min, set back to equilibration at 1% at 90–91 min, and left at 1% for 9 min was used to elute the peptides. Protein quantification was performed using Progenesis QI software version 3.1 (Nonlinear Dynamics, Ltd., Newcastle upon Tyne, UK). All MS/MS data were exported as Mascot generic files (mgf) and searched against the Swiss-Prot Database using MASCOT engine database searching algorithms, version 2.4.0 (www.matrixscience.com (accessed on 25 August 2021)) in Homo sapiens taxonomy. Relative protein abundance was normalized using unique peptides with a MASCOT score >30. Proteins with an ANOVA *p*-value of <0.05 obtained from triplicate runs were considered for subsequent analyses. Log2 fold change >1.2 and *p*-value < 0.05 were used as cut-offs for DEPs.

### 4.8. Bioinformatics Analysis

The profiling data were visualized as a heat map using SRplot (https://www.bioinformatics.com.cn/srplot (accessed on 10 September 2021)). Cluster analysis was performed using the k-means clustering method in STRING (http://string-db.org/ (accessed on 10 September 2021)). The web-based Database Annotation Visualization Integrated Discovery (DAVID) platform was utilized to extract biological explanations regarding the three main Gene Ontology (GO) processes and for pathway analyses. To identify blood-based proteins that reflected tissue-specific biomarkers, all DEGs were converted to proteins using the UniProt database (https://www.uniprot.org/uploadlists/ (accessed on 25 September 2021)). Proteins converted from transcriptome sequencing and DEPs from serum proteomics were integrated to identify overlapping proteins using Venn diagrams (https://bioinfogp.cnb.csic.es/tools/venny/ (accessed on 25 September 2021)).

### 4.9. Western Blotting

The cells were lysed using RIPA buffer (pH 7.4) containing a protease inhibitor cocktail (Millipore, Burlington, MA, USA). For the serum, a dilution of 1:8000 was made using distilled water. The total protein yield was quantified using the Bradford protein assay (Bio-Rad Laboratories). Proteins were separated by 12% sodium dodecyl sulfate-polyacrylamide gel electrophoresis (SDS-PAGE) and transferred onto a nitrocellulose membrane (Bio-Rad Laboratories) at 4 °C overnight. The cell lysate was loaded at 50 µg/well, while the serum dilution was loaded at 5 µg/well. After blocking with 3% bovine serum albumin (Sigma–Aldrich) for 2 h at room temperature, the membranes were probed with primary antibodies at a dilution of 1:20,000 for A1AG1 (Abcam, Cambridge, UK), 1:5000 for FGA (C-7) (Santa Cruz Biotechnology, Inc., Dallas, TX, USA), which specifically recognizes an epitope located between amino acids 581-607 within the internal region of human FGA, 1:20,000 for transferrin (Santa Cruz Biotechnology, Inc.), 1:3000 for β-actin (Cell Signaling Technology, Danvers, MA, USA), and 1:500 for PARP, cleaved PARP, caspase-7, cleaved caspase-7, caspase-3, and cleaved caspase-3 (Cell Signaling Technology) at 4 °C overnight. The membranes were then incubated with horseradish peroxidase-conjugated anti-mouse or anti-rabbit antibodies at a dilution of 1:2000 (Cell Signaling Technology) at room temperature for 2 h. Bands were identified using Clarity Western ECL substrate (Bio-Rad Laboratories). Images were acquired using an ImageQuantTM LAS 4000 digital imaging system (GE Healthcare, Little Chalfont, UK). The relative expression of each protein in the cells was calculated using transferrin as a loading control. The remaining serum albumin after SDS-PAGE was stained with Coomassie blue and used as the loading control.

### 4.10. Plasmid Preparation and Lentiviral shRNA Packaging

*Escherichia coli* with the pGIPZ lentiviral vector carrying each shRNA sequence against the human FGA gene were purchased from GE Healthcare Dharmacon, Inc. (Lafayette, CO, USA). The shRNA sequences were designed based on GenBank accession number KR711790.1 as follows: sequence 1, TGACTTCATCAATCAACCC; sequence 2, TATCAGTGGTAAGTGTTGC; and sequence 3, AACTTGAAGATTTACCACG. Vectors were grown in 2× LB broth containing 100 µg/mL carbenicillin (MedChemExpress, Monmouth Junction, NJ, USA) at 37 °C for 19 h with vigorous shaking. The cell pellets were harvested by centrifugation at 12,000 rpm for 1 min at room temperature. Lentiviral plasmid vectors were purified using the TIANpure Mini Plasmid Kit (TIANGEN Biotech Co., Ltd., Beijing, China). The identity of the plasmid vectors was confirmed by Sanger sequencing using the following sequence: GCATTAAAGCAGCGTATC. Purified plasmid vectors were co-transfected using a Trans-Lentiviral shRNA Packing Kit (GE Healthcare Dharmacon, Inc.), according to the manufacturer’s protocol. Briefly, HEK-293T cells (1.2 × 10^6^ cells/well) were seeded in a 6-well plate in Eagle’s minimum essential medium (EMEM; Gibco, Thermo Fisher Scientific) supplemented with 10% fetal bovine serum (FBS; Gibco) overnight. The next day, 6 µg of the pGIPZ lentiviral vector and 4.3 µL of Trans-Lentiviral Packing Mix were mixed in a total volume of 135 μL with sterile water, followed by the addition of 15 μL of CaCl_2_ and 150 μL of 2× Hanks’ Balanced Salt Solution (HBSS). The transfection mixture was incubated at room temperature for 3 min and then added dropwise to the cells. The cells were incubated at 37 °C with 5% CO_2_ for 16 h, and the medium was replaced with Opti-MEM (Gibco). Green fluorescent protein (GFP) expression was observed under a fluorescence microscope at 24 and 48 h after transfection. The lentiviral supernatant was harvested 48 h after transfection by centrifugation at 2000 rpm and 4 °C for 5 min. Viral particles were concentrated using the Lenti-X™ Concentrator (Takara Bio Inc. Kusatsu, Shiga, Japan) according to the manufacturer’s protocol. The titer of each lentivirus vector was 0.7 × 10^5^ infectious units per mL.

### 4.11. Generation of Lentivirus-Mediated shRNA Interference Targeting FGA

A549 cells (5 × 10^4^ cells/well) were seeded in 24-well plates and incubated overnight. Lentivirus particles at a multiplicity of infection (MOI) of 3 for each vector were mixed with Opti-MEM media containing 8 µg/mL of polybrene in a final volume of 250 µL and then added to the cells. At 6 h post-transduction, an additional 1 mL of Roswell Park Memorial Institute (RPMI) medium (Gibco) with 10% FBS (Gibco) was added and further incubated for 48 h. Stable FGA knockdown cells (shRNA-FGA) were selected by adding 5 µg/mL of puromycin to the media for 1–2 months. Cells were observed under a fluorescence microscope to confirm GFP positivity.

### 4.12. Trypan Blue Dye Exclusion Assay

A549 and shRNA-FGA cells (1 × 10^5^ cells/well) were seeded in 12-well plates and incubated at 37 °C for 24 h. Cells were treated with a combination of gemcitabine and cisplatin at a concentration of 2 µM of each drug. Cells were harvested on days 1, 2, and 3 after drug treatment and washed twice with phosphate-buffered saline (PBS). The cells were resuspended in 100 µL of PBS, and 10 µL of cell suspension was combined with 10 µL of 0.4% Trypan Blue (Gibco). The numbers of viable and dead cells were counted under an inverted microscope (Olympus Corporation, Tokyo, Japan) using a hemocytometer.

### 4.13. MTT (3-(4,5-Dimethylthiazol-2-yl)-2,5-diphenyltetrazolium bromide) Assay

A549 and shRNA-FGA cells were seeded in 96-well plates at a density of 1 × 10^4^ cells/well and incubated at 37 °C for 24 h. The cells were subjected to singlet chemotherapy with either gemcitabine (0.31, 0.62, 1.25, 2.5, 5, 10 and 20 µM) or cisplatin (15.62, 31.25, 62.5, 125, 250, 500 and 1000 µM). Additionally, the cells were treated with doublet chemotherapy, which consisted of a combination of gemcitabine and cisplatin in a range of 0.31, 0.62, 1.25, 2.5, 5, 10, and 20 µM for each drug. Following incubation for 3 days, 5 mg/mL of MTT solution was added to each well and incubated for 2 h at 37 °C. The supernatant was discarded, and dimethyl sulfoxide (Gibco) was added. The optical densities at 550 and 650 nm were measured using a microplate reader (Molecular Devices, San Jose, CA, USA). Cell viability was calculated as the percentage of viable cells. The IC_50_ value, the concentration at which a substance exerts half of its maximal inhibitory effect, was calculated using an IC_50_ online calculator (https://www.aatbio.com/tools/ic50-calculator (accessed on 25 March 2022)). Each condition was tested in duplicate, and three independent MTT assays were performed. The fold-change in resistance was defined as the IC_50_ value of shRNA-FGA divided by the IC_50_ value of the parental cells.

### 4.14. Wound Healing Assay

A549 and shRNA-FGA cells were seeded at a concentration of 1 × 10^5^ cells/well into 24-well plates. Following overnight incubation at 37 °C, a wound field was made in the cell-monolayer in each well using an SPL Scarscratcher (Bio Laboratories Pte Ltd., Singapore). The suspended cells were washed twice with PBS, and the wounded cells were incubated in RPMI medium containing 1% FBS at 37 °C for 3 days. Images of the wound area were captured at 0 and 3 days using a microscope (Olympus Corporation). Areas in the scratch wound that were not covered by cells were quantified. The closure rate was calculated as the percentage of the area at 0 h. All experiments were performed in duplicate.

### 4.15. Statistical Analysis

R statistical software version 4.1.2 (RStudio, Inc., Boston, MA, USA) was used for statistical analyses. All experiments were performed in duplicate. The clinical characteristics of the patients were presented as numbers and percentages. Serum A1AG1 and FGA levels are presented as mean ± standard error of the mean (SEM). The Shapiro–Wilk normality test was used to determine the distribution. An independent *t*-test was used to evaluate differences in data between groups. The cut-off values for A1AG1 and FGA were obtained from the best coordinates in the receiver operating characteristic (ROC) curve to provide the maximum sensitivity and specificity. Associations between protein expression and clinicopathological variables were analyzed using the Chi-squared test. Statistical analyses were considered significant when the *p*-value was <0.05.

## 5. Conclusions

In summary, the integration of transcriptome and proteome data revealed that A1AG1 and FGA were DEGs and DEPs detectable in both the tissue and blood of patients with lung ADC receiving a combination of cisplatin and gemcitabine regimens. The expression level of FGA was associated with resistance to this regimen. Furthermore, decreased FGA expression in lung ADC cells led to resistance to this regimen. Collectively, these findings suggest that FGA can be classified as a tissue-specific serum protein and may serve as a blood-based biomarker for predicting the response of patients with lung ADC to treatment with platinum and antimetabolite drugs.

## Figures and Tables

**Figure 1 ijms-26-01010-f001:**
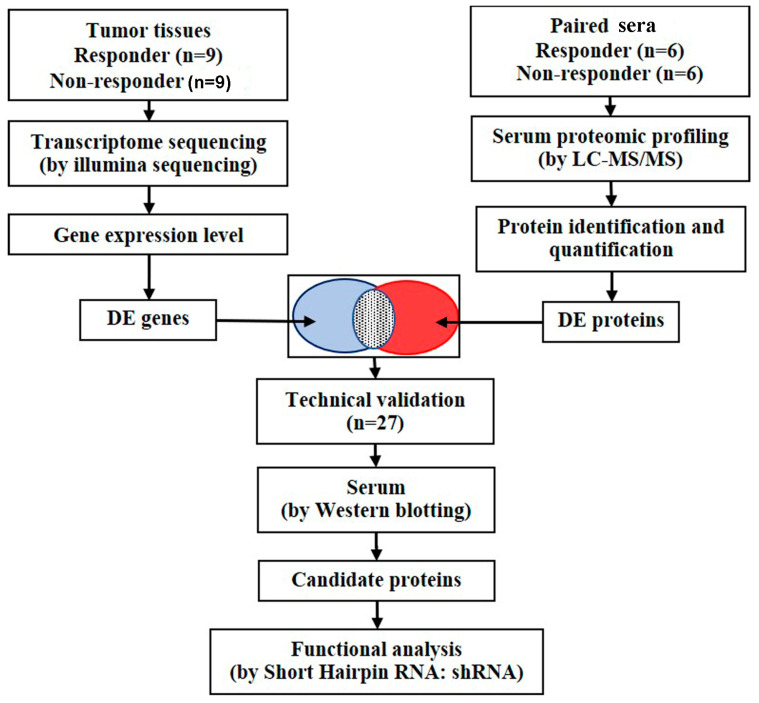
Workflow of study.

**Figure 2 ijms-26-01010-f002:**
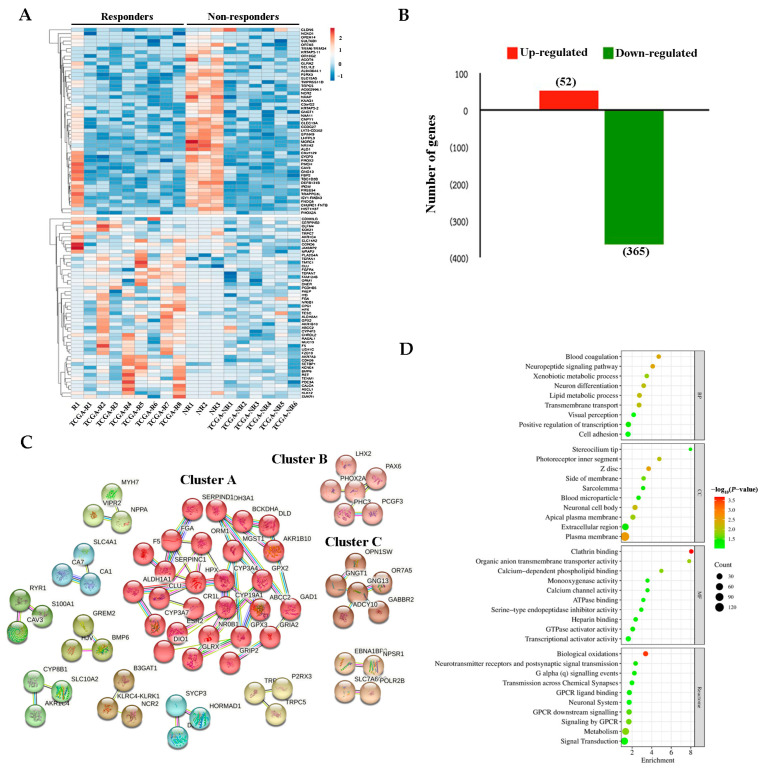
Gene expression profile of chemotherapy responders and non-responders as indicated by transcriptomic analysis. (**A**) Heat map of differentially expressed genes (DEGs). Red indicates high expression and blue indicates low expression. (**B**) Number of upregulated and downregulated genes. (**C**) Cluster analysis of DEGs using STRING. The identified clusters are colored in red (Cluster A), pink (Cluster B), and brown (Cluster C). The solid line indicates connection within the same cluster. Different colors indicate different types of interactions. (Cyan-from curated databases; Pink-experimentally determined; Blue-gene co-occurrence; Khaki-from text mining; Black-coexpression; Light blue-protein homology). (**D**) Gene Ontology (GO) classification and Reactome pathway enrichment analyses of DEGs.

**Figure 3 ijms-26-01010-f003:**
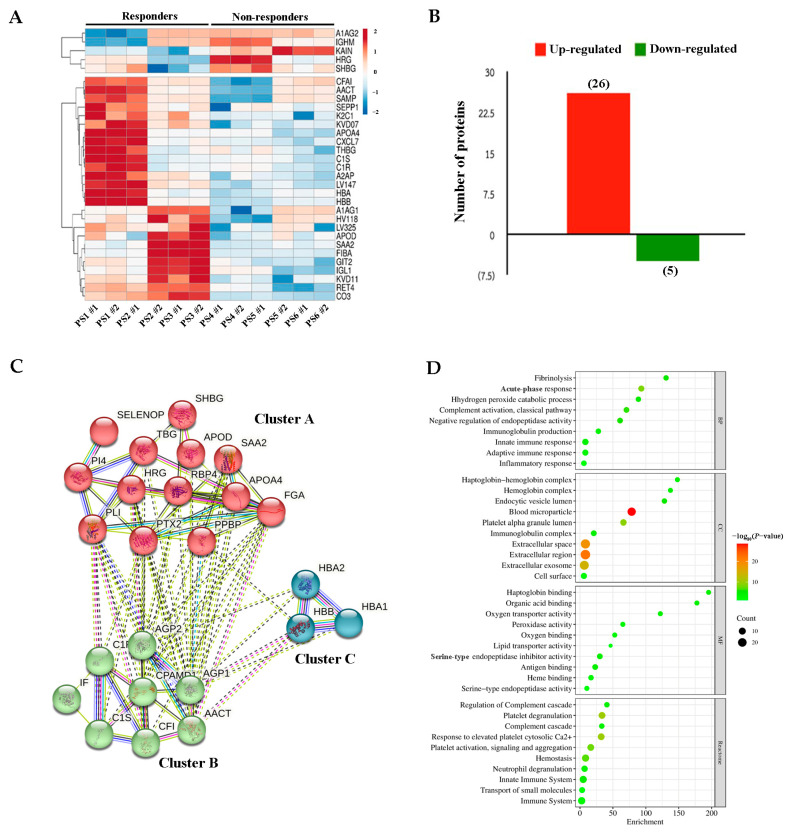
Protein expression profile of chemotherapy responders and non-responders as indicated by proteomic analysis. (**A**) Heat map of differentially expressed proteins (DEPs). Red indicates high expression and blue indicates low expression. (**B**) The number of upregulated and downregulated proteins. (**C**) Cluster analysis of DEPs using STRING. The identified clusters are colored in red (Cluster A), green (Cluster B), and blue (Cluster C). The solid and the dotted lines indicate connection within the same and different cluster respectively. Different colors indicate different types of interactions. (Cyan-from curated databases; Pink-experimentally determined; Blue-gene co-occurrence; Khaki-from text mining; Black-coexpression; Light blue-protein homology). (**D**) Gene Ontology (GO) classification and Reactome pathway enrichment analyses of DEPs.

**Figure 4 ijms-26-01010-f004:**
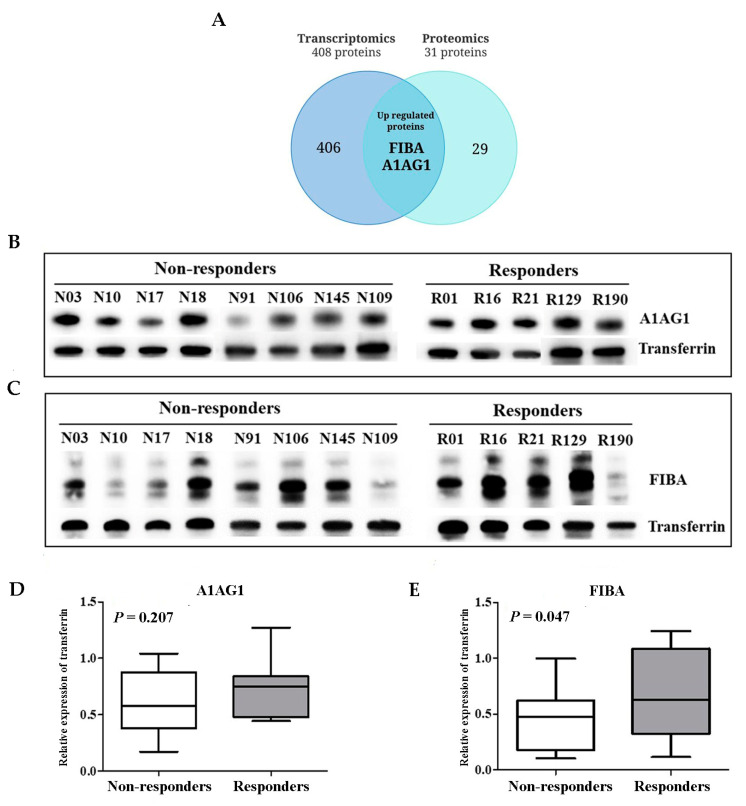
Integrated transcriptomic and proteomic analysis. (**A**) Venn diagram of profiling showing overlap of differentially expressed genes (DEGs) and differentially expressed proteins (DEPs). (**B**,**C**) Serum expression level of A1AG1 and FIBA proteins, respectively, as indicated by Western blotting. (**D**,**E**) Box plot of the relative protein expression of A1AG1 and FIBA, respectively.

**Figure 5 ijms-26-01010-f005:**
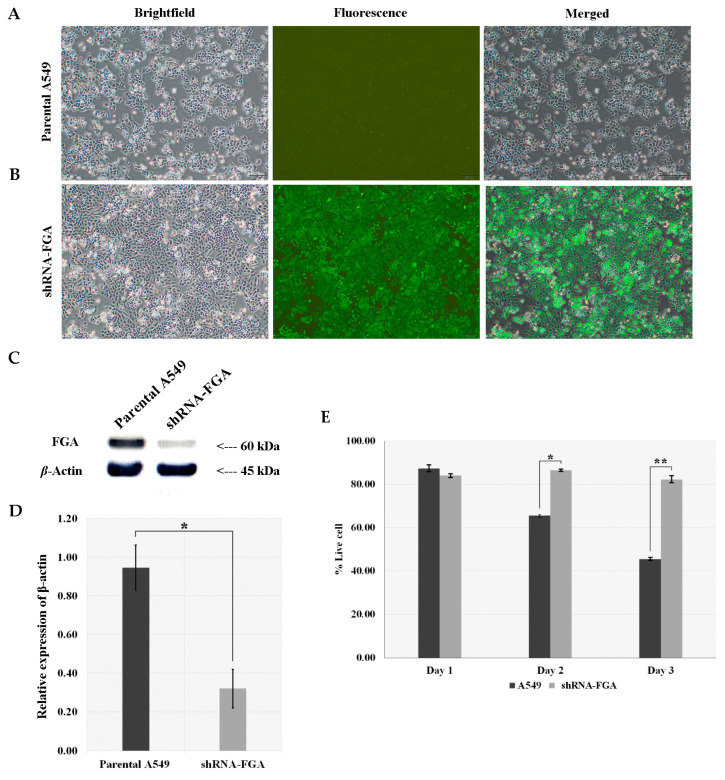
Verification of FGA suppression in A549 cells by lentivirus-mediated shRNA. (**A**) GFP expression images of A549 cells. (**B**) GFP expression images of shRNA-FGA cells showing shRNA delivery efficiency (magnification 20×). (**C**) Band intensities of FGA in A549 and shRNA-FGA cells as indicated by Western blotting. (**D**) Relative expression of FGA in A549 and shRNA-FGA cells. (**E**) Number of viable A549 and shRNA-FGA cells stained by trypan blue reagent after doublet cisplatin and gemcitabine treatment. Comparison group showing significant differential expression using independent *t*-test with * *p* < 0.05; ** *p* < 0.001.

**Figure 6 ijms-26-01010-f006:**
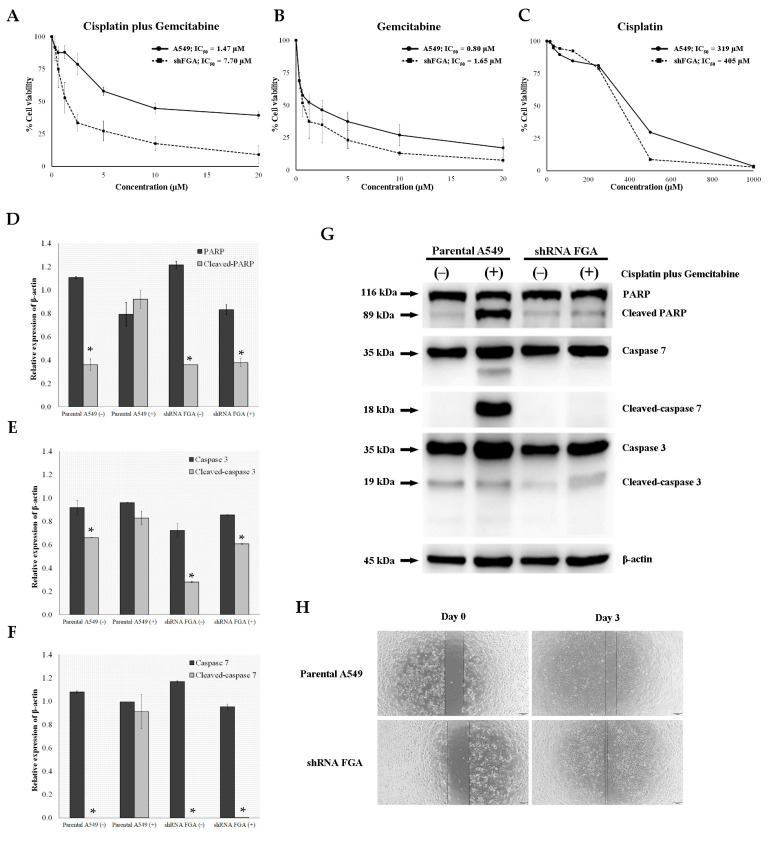
Effect of FGA suppression after lentivirus-mediated shRNA. Determination of IC_50_ values for combination of cisplatin and gemcitabine (**A**); gemcitabine alone (**B**); and cisplatin alone (**C**) as indicated by MTT assay. Relative expression of apoptosis-related proteins with loading control, including PARP and cleaved-PARP (**D**); caspase-3 and cleaved caspase-3 (**E**); and caspase-7 and cleaved caspase-7 (**F**). (**G**) Band intensities of apoptosis-related proteins A549 or shRNA-FGA cells untreated (−) or treated (+) with cisplatin and gemcitabine as indicated by Western blotting. (**H**) Bright-field images showing cell migration of A549 or shRNA-FGA cells in wound healing assay (magnification 20×). Comparison group showing significant differential expression using independent *t*-test with * *p* < 0.05.

**Table 1 ijms-26-01010-t001:** Clinicopathological characteristics of 27 patients.

Variables	Category	Number (%)
Sex		
	Female	13 (48.1)
	Male	14 (51.9)
Age (years)		
	<60	13 (48.1)
	≥60	14 (51.9)
Smoke		
	No	17 (63)
	Yes	10 (37)
Drink		
	No	19 (70.4)
	Yes	8 (29.6)
Stage		
	3	3 (11.1)
	4	24 (88.9)
Response		
	CR	1 (3.7)
	PR	11 (40.7)
	SD	9 (33.3)
	PD	6 (22.2)

Abbreviations: CR, complete response; PR, partial response; SD, stable disease; PD, progressive disease.

**Table 2 ijms-26-01010-t002:** List of top 30 differentially expressed genes (DEGs) as indicated by transcriptomic analysis.

Gene Symbol	Ensembl ID	Annotated Protein Symbol	log2FC	*p*-Value *	*p*-Adj **
CALCA	ENSG00000110680.11	CALC, CALCA	7.9954	2.04 × 10^−7^	8.61 × 10^−5^
TRPC7	ENSG00000069018.16	TRPM2, TRPC7	7.9069	3.09 × 10^−4^	1.65 × 10^−2^
AKR1B10	ENSG00000198074.8	AK1BA	7.8999	2.15 × 10^−7^	8.83 × 10^−5^
CPS1	ENSG00000021826.13	CPSM	6.5426	2.06 × 10^−8^	1.62 × 10^−5^
ALDH3A1	ENSG00000108602.16	AL3A1	6.4751	6.02 × 10^−8^	3.63 × 10^−5^
OLFM4	ENSG00000102837.6	OLFM4	6.3730	4.74 × 10^−6^	6.94 × 10^−4^
NR0B1	ENSG00000169297.7	NR0B1	5.9363	2.32 × 10^−5^	2.34 × 10^−3^
ASCL1	ENSG00000139352.3	ASCL1	5.8997	1.25 × 10^−5^	1.48 × 10^−3^
SLC14A2	ENSG00000132874.12	UT2	5.8886	2.63 × 10^−6^	4.78 × 10^−4^
MUC13	ENSG00000173702.6	MUC13	5.3260	2.56 × 10^−6^	4.71 × 10^−4^
PAEP	ENSG00000122133.15	PAEP	5.1279	4.68 × 10^−5^	3.93 × 10^−3^
AKR1C4	ENSG00000198610.9	AK1C4	5.0082	1.20 × 10^−4^	7.82 × 10^−3^
USH1C	ENSG00000006611.14	USH1C	4.7127	1.09 × 10^−4^	7.30 × 10^−3^
ABCC2	ENSG00000023839.9	MRP2	4.6668	9.99 × 10^−6^	1.24 × 10^−3^
FZD10	ENSG00000111432.4	FZD10	4.6245	5.58 × 10^−5^	4.42 × 10^−3^
FGA	ENSG00000171560.13	FIBA	4.3066	7.00 × 10^−4^	3.17 × 10^−2^
KLK12	ENSG00000186474.14	KLK12	4.2535	6.15 × 10^−4^	2.90 × 10^−2^
CD300LG	ENSG00000161649.11	CLM9	4.1614	5.25 × 10^−4^	2.55 × 10^−2^
SERPINB3	ENSG00000057149.13	SPB3	4.0722	6.23 × 10^−4^	2.91 × 10^−2^
BMP6	ENSG00000153162.8	BMP6	4.0414	4.29 × 10^−6^	6.65 × 10^−4^
NOXO1	ENSG00000196408.10	NOXO1	−10.9752	2.06 × 10^−4^	1.22 × 10^−2^
NCR2	ENSG00000267261.4	NCTR2	−10.5805	1.38 × 10^−11^	7.97 × 10^−8^
PRSS54	ENSG00000096264.12	PRS54	−9.7211	1.96 × 10^−9^	2.60 × 10^−6^
LHFPL3	ENSG00000279018.1	LHPL3	−9.4202	5.39 × 10^−9^	5.54 × 10^−6^
NR1H2	ENSG00000103023.10	NR1H2	−9.0295	2.00 × 10^−14^	3.46 × 10^−10^
FNDC8	ENSG00000268643.1	FNDC8	−8.7288	5.10 × 10^−11^	2.20 × 10^−7^
CLEC19A	ENSG00000187416.10	CL19A	−8.3502	5.45 × 10^−9^	5.54 × 10^−6^
HIST1H3F	ENSG00000131408.12	H31	−8.1709	3.31 × 10^−7^	1.19 × 10^−4^
NRAP	ENSG00000280778.1	NRAP	−8.1458	1.30 × 10^−9^	2.17 × 10^−6^
TRIM6-TRIM34	ENSG00000251357.4	B2RNG4	−8.1102	2.62 × 10^−5^	2.53 × 10^−3^

Abbreviations: FC, fold change; * significant differences identified by an independent *t*-test (*p* < 0.05); ** significant differences for multiple testing identified by the False Discovery Rate (FDR) (*p* < 0.05); (+) denotes upregulation; and (−) denotes down-regulation.

**Table 3 ijms-26-01010-t003:** List of differentially expressed proteins (DEPs) as indicated by proteomic analysis.

Protein Symbol	Accession	Protein Name	Unique Peptides	log2FC	*p*-Value *
SAA2	SAA2_HUMAN	Serum amyloid A-2 protein	2	3.0683	3.04 × 10^−2^
RET4	RET4_HUMAN	Retinol-binding protein 4	3	2.2992	3.81 × 10^−5^
LV147	LV147_HUMAN	Immunoglobulin lambda variable 1–47	1	2.1425	5.03 × 10^−3^
K2C1	K2C1_HUMAN	Keratin, type II cytoskeletal 1	1	2.1402	6.41 × 10^−3^
HBA	HBA_HUMAN	Hemoglobin subunit alpha	1	2.0593	2.96 × 10^−2^
A1AG1	A1AG1_HUMAN	Alpha-1-acid glycoprotein 1	3	1.9118	4.96 × 10^−2^
FIBA	FIBA_HUMAN	Fibrinogen alpha chain	2	1.8921	2.34 × 10^−2^
HBB	HBB_HUMAN	Hemoglobin subunit beta	7	1.8515	1.44 × 10^−2^
CO3	CO3_HUMAN	Complement C3	6	1.8333	3.04 × 10^−4^
GIT2	GIT2_HUMAN	ARF GTPase-activating protein GIT2	1	1.7255	5.27 × 10^−3^
SEPP1	SEPP1_HUMAN	Selenoprotein P	1	1.6813	2.24 × 10^−2^
APOA4	APOA4_HUMAN	Apolipoprotein A-IV	11	1.6533	1.85 × 10^−2^
CXCL7	CXCL7_HUMAN	Platelet basic protein	3	1.5721	1.82 × 10^−2^
KVD07	KVD07_HUMAN	Immunoglobulin kappa variable 3D-7	1	1.5575	1.08 × 10^−3^
IGL1	IGL1_HUMAN	Immunoglobulin lambda-1 light chain	2	1.3985	1.22 × 10^−2^
THBG	THBG_HUMAN	Thyroxine-binding globulin	4	1.3594	4.55 × 10^−2^
KVD11	KVD11_HUMAN	Immunoglobulin kappa variable 3D-11	1	1.3542	2.21 × 10^−3^
AACT	AACT_HUMAN	Alpha-1-antichymotrypsin	11	1.3511	1.04 × 10^−2^
APOD	APOD_HUMAN	Apolipoprotein D	1	1.3155	2.38 × 10^−2^
A2AP	A2AP_HUMAN	Alpha-2-antiplasmin	4	1.2936	3.82 × 10^−3^
SAMP	SAMP_HUMAN	Serum amyloid P-component	5	1.2844	7.54 × 10^−3^
LV325	LV325_HUMAN	Immunoglobulin lambda variable 3–25	2	1.2722	6.86 × 10^−3^
C1S	C1S_HUMAN	Complement C1s subcomponent	5	1.2685	4.35 × 10^−2^
C1R	C1R_HUMAN	Complement C1r subcomponent	8	1.2652	3.95 × 10^−2^
HV118	HV118_HUMAN	Immunoglobulin heavy variable 1–18	1	1.2640	1.90 × 10^−2^
CFAI	CFAI_HUMAN	Complement factor I	7	1.2558	1.13 × 10^−2^
KAIN	KAIN_HUMAN	Kallistatin	2	−2.0818	9.60 × 10^−4^
A1AG2	A1AG2_HUMAN	Alpha-1-acid glycoprotein 2	1	−1.7496	4.82 × 10^−2^
HRG	HRG_HUMAN	Histidine-rich glycoprotein	7	−1.6026	3.59 × 10^−2^
SHBG	SHBG_HUMAN	Sex hormone-binding globulin	1	−1.5059	2.96 × 10^−2^
IGHM	IGHM_HUMAN	Immunoglobulin heavy constant mu	7	−1.3817	4.62 × 10^−2^

Abbreviations: FC, fold change; * significant differences identified by an independent *t*-test (*p* < 0.05); (+) denotes upregulation; and (−) denotes down-regulation.

**Table 4 ijms-26-01010-t004:** Association of clinicopathological variables with protein expression.

Variables	A1AG1 Expression		*p*-Value	FIBA Expression		*p*-Value
	High (%)	Low (%)		High (%)	Low (%)	
Sex			1.000			1.000
Female	10 (76.9)	3 (23.1)		3 (23.1)	10 (76.9)	
Male	10 (71.4)	4 (28.6)		4 (28.6)	10 (71.4)	
Age, years			0.385			0.678
<60	11 (84.6)	2 (15.4)		4 (30.8)	9 (69.2)	
≥60	9 (64.3)	5 (35.7)		3 (21.4)	11 (78.6)	
Smoking			1.000			0.365
No	13 (76.5)	4 (23.5)		3 (17.6)	14 (82.4)	
Yes	7 (70.0)	3 (30)		4 (40.0)	6 (60.0)	
Drinking			1.000			0.011 *
No	14 (73.7)	5 (26.3)		2 (10.5)	17 (89.5)	
Yes	6 (75.0)	2 (25.0)		5 (62.5)	3 (37.5)	
Chemotherapy response			0.018 *			0.029 *
Response	10 (100.0)	0 (0.0)		5 (50.0)	5 (50.0)	
Non-response	10 (58.8)	7 (41.2)		2 (11.8)	15 (88.2)	

Abbreviations: * comparison group showing significant differential expression using the Chi-squared test.

## Data Availability

Data are contained within the article and Appendix A.

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
