# Peer review of "Fibrinogen Alpha Chain as a Potential Serum Biomarker for Predicting Response to Cisplatin and Gemcitabine Doublet Chemotherapy in Lung Adenocarcinoma: Integrative Transcriptome and Proteome Analyses"

_ijms, 2025, doi:10.3390/ijms26031010_

Round 1

Reviewer 1 Report

Comments and Suggestions for Authors

Major comments:

The authors performed biomarker research by combining with transcriptomic and proteomics analysis. The study is well organized and they found serum FGA as a predictive marker for the chemotherapy. They performed a validation study by Western blot analysis using serum samples. The authors should perform a validation study with ELISA so that they can provide quantitative data more accurately. This is very important to convince people for clinical application using FGA as a biomarker.

Abstract

“ADC” needs spell out

Figure1.

In the workflow chart, the authors mentioned technical validation process. However, it is difficult to understand how to perform it. The authors need to describe the technical validation process in the result section of first paragraph.

L183: Is four, correct?

Using our four established sequences (responders = 1 and non-responders = 4) and

Fig6 G

Please add explanation of (+) in the figure legend.

Discussion

Data on Figure 3 suggested importance of coagulation cascades in responder patients, authors add this point to support FGA as a predictive biomarker.

Method

L.616

Does the antibody (FGA (Santa Cruz Biotechnology, Inc., Dallas, Texas, USA)) recognize a-chain specific or recognize other sub-types? Please add this information.

Author Response

Comment 1: The authors performed biomarker research by combining with transcriptomic and proteomics analysis. The study is well organized and they found serum FGA as a predictive marker for the chemotherapy. They performed a validation study by Western blot analysis using serum samples. The authors should perform a validation study with ELISA so that they can provide quantitative data more accurately. This is very important to convince people for clinical application using FGA as a biomarker.

Response 1: I totally agree with the reviewer’s comment. Employing ELISA to quantify FGA levels in serum is significantly more precise than western blotting and is also more convenient for physicians in routine practice. In our present study, we employed western blot to validate the high-throughput technique. This approach can yield semi-quantitative results that are as reliable as ELISA. However, in our next study, we intend to quantify FGA in the serum of a larger patient cohort using the ELISA technique for prospective clinical advantage. Additionally, we plan to utilize the ROC curve to assess the effectiveness of its application as a predictive biomarker. 

Comment 2: Abstract “ADC” needs spell out

Response 2: The suggestion has been included in the abstract section. I have corrected it from ADC to "adenocarcinoma".

Comment 3: Figure1. In the workflow chart, the authors mentioned technical validation process. However, it is difficult to understand how to perform it. The authors need to describe the technical validation process in the result section of first paragraph.

Response 3: The suggestion has been included in the result section. “Overlapping proteins were technically validated through western blotting in serum samples.”

Comment 4: L183: Is four, correct? Using our four established sequences (responders = 1 and non-responders = 4) and

Response 4: I made an error in my original text, and I have since corrected it on L183 from non-responders = 4 to non-responders = 3.

Comment 5: Fig6 G Please add explanation of (+) in the figure legend.

Response 5: The suggestion has been included in the legend of Figure 6 and I have also provided additional detail in the figure. “(G) Band intensities of apoptosis-related proteins A549 or shRNA-FGA cells untreated (-) or treated (+) with cisplatin and gemcitabine by western blotting.”

Comment 6: Discussion Data on Figure 3 suggested importance of coagulation cascades in responder patients, authors add this point to support FGA as a predictive biomarker.

Response 6: The suggestion has been included in the discussion section. “It is a vital protein that is involved in the coagulation of blood in humans [24]. This was supported by our findings that FGA was enriched in the complement and coagulation cascades, as determined by cluster analysis of both DEGs and DEPs.” 

Comment 7: Method L.616 Does the antibody (FGA (Santa Cruz Biotechnology, Inc., Dallas, Texas, USA)) recognize a-chain specific or recognize other sub-types? Please add this information.

Response 7:  The suggestion has been included in the method section (4.9 Western Blotting) “1:5,000 for FGA (C-7) (Santa Cruz Biotechnology, Inc., Dallas, Texas, USA), which specifically recognizes an epitope located between amino acids 581-607 within the internal region of human FGA”

Reviewer 2 Report

Comments and Suggestions for Authors

The author of the manuscript identified Fibrinogen Alpha Chain as a potential serum biomarker for predicting response to Cisplatin and Gemcitabine doublet chemotherapy in lung adenocarcinoma. I have the following questions: 

1.How many samples were ultimately tested in the transcriptome and proteome respectively? There is a significant discrepancy between the description from lines 183-186 and the legend of Figure 1. What were the criteria for selecting the samples for sequencing, specifically the distinction between responder and non-responder based on CR, PR, SD, or PD? Was there an error in the text or the number in line 183? "four" isn’t equal "1+4".

2. What were the criteria for selecting the data from the TCGA dataset in Figure 2A? In line 191, does 'Clusters B and C consisted of six genes' mean that each of B and C contains six genes, or do they together contain six genes in total? What do the * and ** in Table 2 represent?

3. The format represented by numbers should be unified, whether in text or numerical form. Line 240,242,243.

4. Is the heatmap in Figure 3A created after selecting the differentially expressed proteins between responders and non-responders?  FIBA seems to have less obvious differences, PS1#1, PS#2, PS#3 and PS4, PS5, PS6? additional samples are required to perform Western blot validation to ensure the reliability of the key conclusion.

5.Table 4: 4 (28.6) 10 (28.6). not 100%; How were the gene expression levels in the 27 patient samples detected? Was it through transcriptome analysis, proteome analysis, Western blot, or another method?

6. The author intends to use FGA as a biomarker and should plot a ROC curve to test and evaluate the performance of the protein.

Author Response

The author of the manuscript identified Fibrinogen Alpha Chain as a potential serum biomarker for predicting response to Cisplatin and Gemcitabine doublet chemotherapy in lung adenocarcinoma. I have the following questions: 

Comment 1: How many samples were ultimately tested in the transcriptome and proteome respectively? There is a significant discrepancy between the description from lines 183-186 and the legend of Figure 1. What were the criteria for selecting the samples for sequencing, specifically the distinction between responder and non-responder based on CR, PR, SD, or PD? Was there an error in the text or the number in line 183? "four" isn’t equal "1+4".

Response 1: 

  • The sample numbers in Figure 1 have been revised (transcriptomic; responders = 9 and non-responders = 9), and the correction on line 183 has been made from non-responders = 4 to non-responders = 3.
  • The criteria for sample selection for sequencing of both transcriptome and proteome have been explained previously in the methods section (4.1 and 4.2). Response assessment involved dichotomizing patients into responders (complete response/partial response) and non-responders (stable disease/progressive disease) according to RECIST criteria, as previously detailed in the methods section (4.3).
  • I made an error in my original text, and I have since corrected it on L183 from non-responders = 4 to non-responders = 3.

Comment 2: What were the criteria for selecting the data from the TCGA dataset in Figure 2A? In line 191, does 'Clusters B and C consisted of six genes' mean that each of B and C contains six genes, or do they together contain six genes in total? What do the * and ** in Table 2 represent?

Response 2:

  • The criteria for data selection from the TCGA dataset have been detailed previously in the methods section (4.2).
  • Line 191 indicates that both cluster B and cluster C contain six genes each. The corrections regarding this point have been made in the result section (2.2). “Cluster B and C each consisted of 6 genes associated with regionalization, interneuron differentiation, and the suppression of voltage-gated channels”
  • the * and ** in Table 2 indicate significant differences identified using an independent t-test and the False Discovery Rate (FDR), respectively.  Corrections have been made for clarity in the abbreviations of Table 2 and 3. Table 2: “Abbreviations: FC, fold change; * significant differences identified by an independent t-test (p < 0.05); ** significant differences for multiple testing identified by the False Discovery Rate (FDR) (p < 0.05); (+) denotes upregulation; and (–) denotes down-regulation. Table 3: “Abbreviations: FC, fold change; * significant differences identified by an independent t-test (p < 0.05); (+) denotes upregulation; and (–) denotes down-regulation.  

Comment 3: The format represented by numbers should be unified, whether in text or numerical form. Line 240,242,243.

Response 3:  I have corrected the numerical format throughout the manuscript.

Comment 4: Is the heatmap in Figure 3A created after selecting the differentially expressed proteins between responders and non-responders?  FIBA seems to have less obvious differences, PS1#1, PS#2, PS#3 and PS4, PS5, PS6? additional samples are required to perform Western blot validation to ensure the reliability of the key conclusion.

Response 4:

  • Figure 3A illustrates a cluster analysis conducted between two groups: responders and non-responders, after selecting the differentially expressed proteins. The responders included 3 pooled samples in duplicate from 6 samples: PS1 #1, PS1 #2, PS2 #1, PS2 #2, PS3 #1, and PS3 #2. The non-responders included 3 pooled samples in duplicate from 6 samples: PS4 #1, PS4 #2, PS5 #1, PS5 #2, PS6 #1, and PS6 #2.
  • The expression of FIBA (or FGA) in responders was significantly higher than in non-responders, as indicated by color coding, with red representing high expression and blue representing low expression. This result was obtained through a proteomic technique using 12 serum samples. For the subsequent technical validation via western blotting, additional serum samples in a total of 27 samples have been added to assess the reliability of FGA.   

Comment 5: Table 4: 4 (28.6) 10 (28.6). not 100%; How were the gene expression levels in the 27 patient samples detected? Was it through transcriptome analysis, proteome analysis, Western blot, or another method?

Response 5:

  • An error in the original text has been corrected in Table 4 from 10 (28.6) to 10 (71.4).
  • The protein expression levels were quantified using western blotting, and additional clarification has been provided in the results section (2.5). “The serum level of A1AG1 and FIBA proteins, as determined by western blotting, established cut-off values of ≤0.424 for A1AG1 and ≤0.776 for FGA at the lower value, and >0.424 for A1AG1 and >0.776 for FGA at the upper value.”

Comment 6: The author intends to use FGA as a biomarker and should plot a ROC curve to test and evaluate the performance of the protein.

Response 6: I totally agree with the reviewer’s comment. In our next study, we intend to quantify FGA in the serum of a larger patient cohort using the ELISA technique for prospective clinical advantage. Additionally, we plan to utilize the ROC curve to assess the effectiveness of its application as a predictive biomarker. 

Round 2

Reviewer 1 Report

Comments and Suggestions for Authors

Thank you for your resubmission on the revised version. I understand the authors plan to conduct more precise validation study using ELISA so that they can judge to use the marker in clinical practice. I recommend including the limitation of FGA as a biomarker in clincail practice so far in discussion part. 

Author Response

Comment 1:

Thank you for your resubmission on the revised version. I understand the authors plan to conduct more precise validation study using ELISA so that they can judge to use the marker in clinical practice. I recommend including the limitation of FGA as a biomarker in clincail practice so far in discussion part.

Response 1:

I appreciate your recommendation. The limitations of this study have been included in the discussion section.

“Nonetheless, the current investigation was constrained by the restricted number of clinical samples, including both tissues and serum, which may hinder the assessment of putative biomarkers for FGA. Consequently, further clinical studies with larger sample sizes are required to confirm the role of FGA as a predictive biomarker in this context.”

Reviewer 2 Report

Comments and Suggestions for Authors

I have an additional question. Could you specify the exact number of samples for the transcriptome sequencing and  LC-MS/MS? Why were only 4 sequencing samples chosen in fig2A?

Author Response

Comment 1:

I have an additional question. Could you specify the exact number of samples for the transcriptome sequencing and  LC-MS/MS? Why were only 4 sequencing samples chosen in fig2A?

Response 1:

  • For transcriptomic sequencing, we intended to use tumor tissues from 18 patients, comprising 9 responders and 9 non-responders. However, we found that only 4 samples achieved the minimum requirements (Table S1). Because of the insufficient quality of RNA in our tissue samples, we obtained 14 tissue-based transcriptome datasets from TCGA database. I have already addressed this issue in the result section (2.1 and 2.2).
  • For proteomic profiling, 12 serum samples (responders = 6 and non-responders = 6) were chosen based on sex–age matching. Two individual serum samples in each group were pooled. Consequently, the responders included 3 pooled samples in duplicate from 6 samples: PS1 #1, PS1 #2, PS2 #1, PS2 #2, PS3 #1, and PS3 #2. The non-responders included 3 pooled samples in duplicate from 6 samples: PS4 #1, PS4 #2, PS5 #1, PS5 #2, PS6 #1, and PS6 #2. The corrections regarding this point have been implemented in the result section (2.1 and 2.3).
  • In addition, to enhance clarity, I have included additional information regarding a number of serum samples that were employed for technical validation in the result section (2.4).